

# Collateral transgression of planetary boundaries due to climate engineering by terrestrial carbon dioxide removal

Vera Heck [1,3], Jonathan F. Donges [1,2], and Wolfgang Lucht [1,3,4]

[1]Earth System Analysis, Potsdam Institute for Climate Impact Research, Telegraphenberg A62, 14473 Potsdam, Germany
[2]Stockholm Resilience Centre, Stockholm University, Kräftriket 2B, 114 19 Stockholm, Sweden
[3]Department of Geography, Humboldt-Universität zu Berlin, Unter den Linden 6, 10099 Berlin, Germany
[4]Integrative Research Institute on Transformations of Human-Environment Systems, Unter den Linden 6, 10099 Berlin, Germany

*Correspondence to:* Vera Heck (heck@pik-potsdam.de)

**Abstract.** The planetary boundaries framework as proposed by Rockström et al. (2009) provides guidelines for defining thresholds in environmental variables. Their transgression is likely to result in a shift in Earth system functioning away from the relatively stable Holocene state. As the climate change boundary is already transgressed, several climate engineering methods are discussed, aiming

at a reduction of atmospheric carbon concentrations to control the Earth's energy balance. Terrestrial carbon dioxide removal (tCDR) via afforestation or bioenergy production with carbon capture and storage are part of most climate change mitigation scenarios that limit global warming to less than 2°C.

    We analyse the co-evolutionary interaction of societal interventions via tCDR and the natural

dynamics of the Earth's carbon cycle. Applying a conceptual modelling framework, we analyse how societal monitoring and management of atmospheric $CO_2$ concentrations with the aim of staying within a 'safe' level of global warming might influence the state of the Earth system with respect to other carbon-related planetary boundaries.

    Within the scope of our approach, we show that societal management of atmospheric carbon via

tCDR can lead to a transgression of the planetary boundaries of land system change and ocean acidification. Our analysis indicates that the opportunities to remain in a desirable region within carbon-related planetary boundaries depend critically on the sensitivity and strength of the tCDR management system, as well as underlying emission pathways. While tCDR has the potential to ensure the Earth system's persistence within a carbon safe operating space under low emission path-

ways, this potential decreases rapidly for medium to high emission pathways.





## 1 Introduction

Rockström et al. (2009) introduced the concept of a safe operating space (SOS) for humanity, delineated by nine global planetary boundaries, some of which take into account the existence of tipping points or nonlinear thresholds in the Earth system (Lenton et al., 2008; Schellnhuber, 2009; Kriegler et al., 2009) and may frame sustainable development. Particularly, the state of the Earth system with respect to climate change has received strong political attention, as atmospheric carbon concentrations have already entered the uncertainty zone of the planetary boundary of climate change, set at an atmospheric $CO_2$ concentration of 350 ppmv to 450 ppmv (Steffen et al., 2015).

The Paris climate agreement (UNFCCC, 2015) aims at limiting global temperature increase to well below 2°C above pre-industrial levels, while currently greenhouse gas emissions are still growing. Fuss et al. (2014) have highlighted that more than 85 % of IPCC scenarios that are consistent with the 2° goal require net negative emissions before 2100. Particularly, terrestrial carbon dioxide removal (tCDR) via afforestation or large-scale cultivation of biomass plantations for the purpose of bioenergy production has been included in recent IPCC scenarios (Vuuren et al., 2011; Kirtman et al., 2013). Furthermore, tCDR has been proposed as a climate engineering (CE) method that could be applied in case global efforts in mitigating anthropogenic greenhouse gas emissions fail to prevent dangerous climate change (Caldeira and Keith, 2010).

In the context of the SOS framework, tCDR could on the one hand extract carbon from the atmosphere via the natural process of photosynthesis (Shepherd et al., 2009). If the carbon accumulated in biomass is harvested and stored in deep reservoirs or used for bioenergy production in combination with carbon capture and storage (Caldeira et al., 2013), further transgression of the climate change boundary and initial transgression of the ocean acidification boundary could be prevented. On the other hand, tCDR is likely to have unintended impacts on other Earth system components besides atmospheric carbon concentrations that is mediated by the global cycles of carbon, water and other biogeochemical compounds (Vaughan and Lenton, 2011). For example, large-scale biomass plantations would require substantial amounts of fertilizer, irrigation water and land area, driving the Earth system closer to the planetary boundaries for biogeochemical flows, freshwater use and land system change, respectively (Heck et al., 2016).

Social and political actions are important drivers of tCDR. The willingness to engage in CE or mitigation is based on monitoring of the climate system and can be expected to increase as the climate system approaches the normatively assigned climate change boundary. A holistic assessment and systemic understanding of CE therefore requires an analysis of the social and ecological co-evolutionary system.

A dynamic integration of complex interactions between the social and ecological components of the Earth system to simulate in detail the co-evolution of societies and the environment is currently unfeasible due to fundamental conceptual problems and high computational demands on both modelling sides (Van Vuuren et al., 2012; van Vuuren et al., 2015). An emerging field of low-complexity





models explores new pathways for understanding social-ecological Earth system dynamics (e.g. Brander and Taylor, 1998; Kellie-Smith and Cox, 2011; Jarvis et al., 2012; Anderies et al., 2013; Motesharrei et al., 2014). For example, first simulation approaches have been reported using such conceptual models to simulate the interaction between human climate monitoring and societal action in the form of transitions to renewable energy (Jarvis et al., 2012) or climate engineering (MacMartin et al., 2013) While not aiming for realism in their quantitative evaluations, the low complexity of such conceptual models allows to understand the structure and effects of dominating feedbacks and their leading interactions, which are otherwise often hidden in the complexity of state-of-the-art full complexity Earth system models.

In this paper, we provide a conceptual but systematic analysis of the nonlinear system response to using tCDR for steering the Earth system within the SOS defined by planetary boundaries as quantified by Rockström et al. (2009) and Steffen et al. (2015). Specifically, we analyse how the trade-offs between tCDR and other planetary boundaries depend on the achievable rate and threshold of tCDR implementation; and whether particular combinations of climate and management parametrisations can safeguard a persistence within the SOS. As a starting point, we focus on a subset of the nine proposed planetary boundaries that are most important in the context of tCDR. These are the carbon-related boundaries on climate change, ocean acidification and land-system change.

We utilise a conceptual model of the carbon cycle and expand it to explore feedbacks within and between societal and ecological spheres, while being sufficiently simple to permit an analysis of its state and parameter spaces in the form of constrained stability analysis similar to van Kan et al. (2016). We do not aim to provide a quantitative assessment because in this exploratory study we choose to use a computationally efficient conceptual model to shed light onto the qualitative structure of co-evolutionary dynamics. The approach proposed here can be transferred to models of higher complexity to the extent that this is computationally feasible.

This paper is structured as follows: following the introduction (Sect. 1) we present a co-evolutionary model of societal monitoring and tCDR intervention in the Earth's carbon cycle and related parameter calibration procedures (Sect. 2). Subsequently, we present and discuss our results (Sect. 3) and finish with conclusions (Sect. 4).

## 2 Methods

In social-ecological systems modelling, societal influences and ecological responses are recognized as equally important (Berkes et al., 2000). Therefore, it can be considered essential that representations of social and ecological systems are of the same order of complexity. Increasing complexity of only one model component would not increase the accuracy of information generated by the full coupled model, but would greatly increase computational demand. In view of our objective, we require a sufficiently simple model that conceptually captures the most important processes of global



carbon dynamics with respect to planetary boundaries, as well as a stylised societal management
loop consisting of tCDR interventions and monitoring of the climate system.

### 2.1 Co-evolutionary model of societal monitoring and tCDR intervention in the carbon cycle

The basis of our co-evolutionary model is the conceptual carbon cycle model by Anderies et al.
(2013), developed specifically to enable a bifurcation analysis of carbon-related planetary bound-
aries and their interactions. We modified atmosphere-land interactions for a better representation of
empirically observed Earth system carbon dynamics and extended the model by a stylized social
management loop mimicking the current focus of international policy processes on climate change.
We calibrated the model in order to represent global carbon cycle dynamics consistent with obser-
vational data and simulations from detailed high-resolution Earth system models (Sect. 2.2). In the
following, we provide an overview of the fundamental model equations. A detailed motivation of
the model design is given in Anderies et al. (2013).

The adapted model consists of five interacting carbon pools: land $C_t(t)$, atmosphere $C_a(t)$, upper
ocean $C_m(t)$, geologic fossil reservoirs $C_f(t)$ and a potential CE carbon sink $C_{CE}(t)$ (Fig. 1). All
model equations are summarised in Table 1.

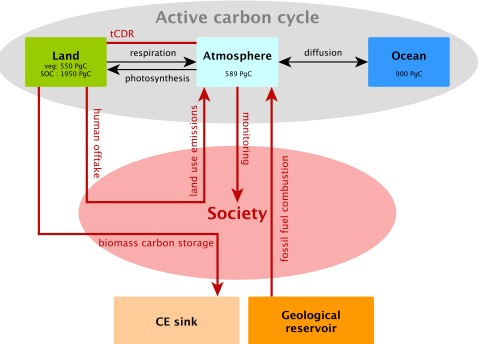

**Figure 1.** Structure of the co-evolutionary model of societal monitoring and tCDR intervention in the carbon
cycle including simulated components of the carbon cycle as well as a societal management loop and their
interactions. Carbon values in the boxes indicate estimates of preindustrial carbon pools in the year 1750 AD
(Batjes, 1996; Ciais et al., 2013)).



| Process | Equation | |
|---|---|---|
| conservation of mass | $C_t(t) + C_a(t) + C_m(t) = C_0 + C_r(t) - C_{CE}(t)$ | (1) |
| fossil carbon release | $\dot{C}_r(t) = r_i C_r(t)(1 - \frac{C_r(t)}{c_{max}})$ | (2) |
| CE carbon storage | $\dot{C}_{CE}(t) = H_{CE}(C_t(t), C_a(t))$ | (3) |
| ocean-atmosphere diffusion | $\dot{C}_m(t) = a_m(C_a(t) - \beta C_m(t))$ | (4) |
| terrestrial carbon flux | $\dot{C}_t(t) = NEP(C_a(t), C_t(t)) - H(C_t(t)) - H_{CE}(C_t(t), C_a(t))$ | (5) |
| net ecosystem productivity | $NEP(C_a(t), C_t(t), T(t)) = r_{tc} \left[ P(T(t)) - R(T(t)) \right] C_t(t) \left[ 1 - \frac{C_t(t)}{K(C_a(t))} \right]$ | (6) |
| terrestrial carbon carrying capacity | $K(C_a(t)) = a_k e^{-b_k C_a(t)} + c_k$ | (7) |
| photosynthesis | $P(T(t)) = a_p T(t)^{b_p} e^{-c_p T(t)}$ | (8) |
| respiration | $R(T(t)) = a_r T(t)^{b_r} e^{-c_r T(t)}$ | (9) |
| temperature | $T(C_a(t)) = a_T C_a(t) + b_T$ | (10) |
| tCDR offtake flux | $H_{CE}(C_t(t), C_a(t)) = \alpha_{CE}(C_a(t)) C_t(t)$ | (11) |
| societal tCDR offtake rate | $\alpha_{CE}(C_a(t)) = \alpha_{max} \left( 1 + \exp(-s_{CE} (C_a(t) - \tilde{C}_a)) \right)^{-1}$ | (12) |
| other human biomass offtake flux | $H(C_t(t)) = \alpha C_t(t)$ | (13) |

**Table 1.** Summary of equations describing the co-evolutionary model of societal monitoring and tCDR intervention in the carbon cycle building upon Anderies et al. (2013).

The co-evolutionary dynamics of the system is determined by Equations (1)–(5). Conservation of mass (Eq. 1) dictates that the active carbon in the system, i.e. the sum of terrestrial, atmospheric and maritime carbon, is equal to the active carbon at preindustrial times ($C_0$) plus carbon released from fossil reservoirs ($C_r(t)$) minus carbon extracted via tCDR ($C_{CE}(t)$) to permanent stores. Fossil carbon release (Eq. 2) is approximated by a logistic function parametrised by the maximum emitted carbon $c_{max}$ and rate of carbon release $r_i$.

The social management loop is motivated by proposals of CE as a management intervention in response to intolerable levels of global warming. It comprises atmospheric carbon monitoring and tCDR action conditional on the proximity to a critical threshold of atmospheric carbon content (Eq. 3). CE action is implemented via a tCDR carbon offtake from terrestrial carbon ($H_{CE}(t)$) and storage in a permanent (geological) sink $C_{CE}$. Carbon offtake for tCDR (Eq. 11) is defined analogous to human offtake for agriculture or land use change (Eq. 13), however, with a dynamic offtake rate $\alpha_{CE}(C_a(t))$ (Eq. 12).

TCDR characteristics are governed by three parameters: (i) implementation threshold ($\tilde{C}_a$) in terms of atmospheric carbon content, representing societal foresightedness, (ii) maximally achievable rate of tCDR ($\alpha_{max}$), a measure of societies' efforts, as well as biogeochemical constraints and (iii) the slope of tCDR implementation ($s_{CE}$), parametrising social and economic implementation capacities. Figure 2 depicts an exemplary tCDR trajectory for constant terrestrial carbon in Eq. (11) for two values of $s_{CE}$. The implementation time can be computed from the slope of tCDR-





implementation by using current increase rates of atmospheric carbon as a conversion factor. With current increase rates of approximately 2 ppmv $a^{-1}$ (Tans and Keeling, 2015), the two depicted values of $s_{CE}$ correspond to tCDR ramp-up times of approximately 20 years and 40 years (from 10 % to 90 % capacity) for $s_{CE} = 0.1$ ppmv$^{-1}$ (solid) and $s_{CE} = 0.05$ ppmv$^{-1}$ (dashed), respectively.

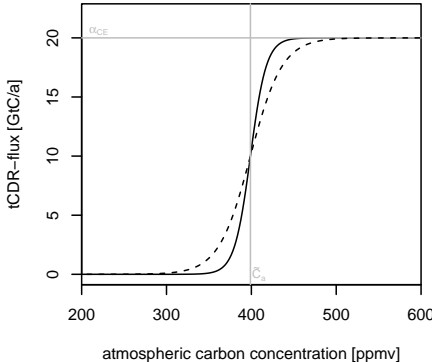

**Figure 2.** Sigmoidal dependence of the tCDR-flux on atmospheric carbon concentrations for two values of the tCDR implementation capacity parameter (slope): $s_{CE} = 0.1$ ppmv$^{-1}$ (solid line) and $s_{CE} = 0.05$ ppmv$^{-1}$ (dashed line). The threshold parameter ($\tilde{C}_a$) is set at 400 ppmv atmospheric carbon concentration and the potentially achievable tCDR-flux is parametrised with $\alpha_{max} = 20$ GtC a$^{-1}$.

The atmosphere and ocean carbon feedback (Eq. 4) is governed by diffusion, depending on the difference between atmospheric and maritime carbon pools.

Land-atmosphere interaction is determined by both ecological and social processes: the net ecosystem productivity (Eq. 6), tCDR offtake (Eq. 11) and other human offtake for agriculture and other land use (Eq. 13), respectively.

Net ecosystem productivity is given by the net carbon flux of photosynthesis (Eq. 8) and respiration (Eq. 9), multiplied with the terrestrial carbon pool and a logistic dampening function which represents competition for space, sunlight, water or nutrients. Both photosynthesis and respiration are continuous functions of global land temperature ($T(t)$, Eq. 10), which in turn depends linearly on atmospheric carbon content. It is important to note that in our model respiration exceeds photosynthesis for higher temperatures (Fig. 3). The state of equilibrium of the terrestrial carbon pool is thus determined by the land surface temperature, as well as the terrestrial carbon carrying capacity (Eq. 7) in the density function. In contrast to Anderies et al. (2013), we implement a dynamic terrestrial carbon carrying capacity as a function of atmospheric carbon content. This is motivated by a number of factors such as CO$_2$ fertilisation and a higher water use efficiency under higher atmospheric carbon concentrations, as well as higher average vegetation density in a warmer world (e.g.


Drake et al., 1997; Keenan et al., 2013). For low atmospheric carbon we assume a rapid increase of terrestrial carbon storage capacity as a function of atmospheric carbon concentration and a saturation of storage capacity for high atmospheric carbon, in line with assessments of coupled carbon-cycle

climate models (Heimann and Reichstein, 2008). The functional relationship in (Eq. 7) follows these constraints for chosen parameter values (Sect. 2.2).

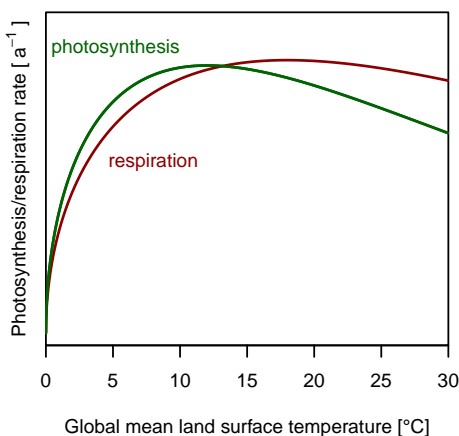

**Figure 3.** Modelled photosynthesis and respiration rates as a function of global mean land surface temperature.

### 2.2    Calibration of model parameters

A sufficiently suitable application of a conceptual model in the context of the planetary boundaries as in Steffen et al. (2015) requires the model's ability to simulate credible transients of global carbon

dynamics. In order to achieve this, we calibrated model parameters to observed carbon fluxes and pools, as well as simulation results of detailed high-resolution Earth system models.

    Because we simulate relative dynamics between the different carbon compartments and do not aim at prognostics of actual time evolution of carbon pools, all carbon fluxes and pools are normalised to the the active carbon at preindustrial times, i.e. the total sum of preindustrial carbon in the year

1750 AD (3989 GtC, Fig. 1). All normalised parameter values are summarised in Table 2.

### 2.2.1    Temperature

The linear relationship between temperature and atmospheric carbon content (Eq. 10) was calibrated to reported long-term global mean surface temperature increase per emitted carbon of 2K/1000GtC (Joos et al., 2013; Gillett et al., 2013). On timescales of a few hundred years, approximately 50 % of

the emitted carbon stays in the atmosphere (Archer et al., 2009; Joos et al., 2013). Thus, global mean


temperature increase rate per atmospheric carbon increase is approximately twice the temperature increase rate of emitted carbon (i.e. 2K/500 GtC in the atmosphere). From this global surface temperature increase rate (2/3 ocean and 1/3 land surface), the global land surface temperature increase can be inferred via the global land/sea warming ratio of approximately 1.6 (Sutton et al., 2007). Thus, we

approximate a global land surface warming rate of 5.3K/1000GtC remaining in the atmosphere. The y-offset ($b_T$ in Eq. 10) was inferred via global land surface temperature anomalies from 1880–2000 (Jones et al., 2012), a global average (1880–2000) land temperature of 8.5 °C (NOAA, 2015) and observed monthly mean $CO_2$ concentrations (Mauna Loa, 1959–2000) (Tans and Keeling, 2015).

### 2.2.2   Ocean-atmosphere dynamics

The carbon solubility in sea water factor ($\beta$) is directly determined by the assumption of pre-industrial equilibrium between upper-ocean and atmospheric carbon ($\dot{C}_m(0) = 0$). From this and a present carbon flux from the atmosphere to the ocean of $\dot{C}_m(t_{tod}) = 2.3$ GtC a$^{-1}$ (Ciais et al., 2013) follows the atmosphere-ocean diffusion coefficient $a_m$.

### 2.2.3   Terrestrial dynamics

Photosynthesis and respiration are calibrated according to temperature relationships reported in the literature. However, literature generally specifies temperature relationships at small temporal and spatial scales in controlled environments, whereas our model equations refer to a global average of day and night-time temperature. Thus, only a rough estimation of the relationship between temperature and photosynthesis/respiration for model calibration is possible. As in Anderies et al. (2013), we

assume a maximum of respiration at a global land surface temperature of 18 °C (supported by Yuan et al. (2011)), determining the ratio of parameters $b_r/c_r = 18$ °C (Fig. 3). We choose a maximum of photosynthesis at 12 °C, incorporating a $CO_2$ fertilisation feedback indirectly via the dependence of temperature on atmospheric carbon ($b_p/c_p = 12$ °C). The amplitudes of photosynthesis and respiration functions ($a_r$ and $a_p$, respectively) are approximated for agreement with carbon fluxes reported

in Ciais et al. (2013). Note that the functional form of carbon fluxes is not decisive for the model dynamics, however, it is important that the curves of photosynthesis and respiration intersect at some temperature limit where ecosystem respiration exceeds photosynthesis. With our parametrisation this is the case at a global mean land surface temperature of approximately 13 °C, which is 4.5 °C warmer than the 20th century average global mean land surface temperature (NOAA, 2015). This is

in line with multi-model assessments in carbon reversal studies (e.g. Heimann and Reichstein, 2008; Friend et al., 2013).

The terrestrial carbon carrying capacity $K(C_a(t))$ in $\dot{C}_t(t)$ determines how much carbon can be accumulated in the terrestrial system. $K(C_a(t))$ was calibrated to represent both, past long term climatic and terrestrial carbon changes (last glacial maximum to Holocene) (Crowley, 1995; François

et al., 1998; Kaplan et al., 2002; Joos et al., 2004) and prognostics of climate change impacts on ter-





restrial carbon storage (Joos et al., 2001; Lucht et al., 2006; Friend et al., 2013), to capture terrestrial changes due to climate variability (Fig. 4).

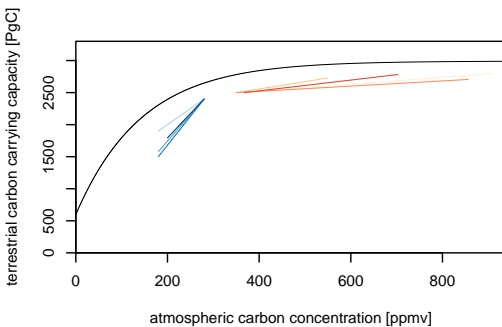

**Figure 4.** Approximated terrestrial carbon carrying capacity (black line). Blue lines represent approximate changes in terrestrial carbon storage published in Crowley (1995); François et al. (1998); Kaplan et al. (2002); Joos et al. (2004). Red lines represent simulated changes in terrestrial carbon storage due to climate change reported by Joos et al. (2001); Lucht et al. (2006); Friend et al. (2013)

Human activities such as fires, deforestation and agricultural land use that affect terrestrial carbon stocks are summarised as human offtake of biomass and are presently estimated at 1.1 GtC $a^{-1}$ (Ciais et al., 2013). With a present terrestrial carbon pool of 2470 GtC follows the human offtake rate $\alpha = H(t_{tod})/C_t(t_{tod})$.

### 2.3 Planetary Boundaries

We use the carbon-related planetary boundaries (climate change, ocean acidification and land system change) to define the desirability of given trajectories of carbon pool evolution. The proposed locations of these boundaries are normalised to match the normalisation of our model. Note that the exact location and normalisation of the boundaries is not decisive for our results because we qualitatively analyse the influence of tCDR management on the existence of desirable trajectories. Slightly different sets of planetary boundaries would not qualitatively change the systemic effects reported in this study.

The planetary boundary for climate change is proposed at 350–450 ppmv $CO_2$ equivalents in the atmosphere (Steffen et al., 2015). Taking the mean, 400 ppmv, the normalised climate change boundary is at 0.21 atmospheric carbon. Ocean acidification is measured via the saturation state of aragonite and its boundary is set at 80 % of the preindustrial average annual global saturation state of aragonite (Steffen et al., 2015). Since chemical processes are not explicitly represented in our model, this measure is not directly transferable to maritime carbon content. However, at the current carbon content (1150 GtC), the saturation state of aragonite is at 84 % of the preindustrial value (Guinotte





and Fabry, 2008). We estimate the normalised ocean acidification boundary at 0.31, slightly higher than the current value (0.29). The land system change boundary is defined in terms of the amount of remaining forest cover and has been specified as 75 % of global forest cover remaining (Steffen et al., 2015). As an estimate, we translate deforestation into carbon content by measuring the loss of vegetation carbon with deforestation. With vegetation carbon of 550 GtC (Ciais et al., 2013), we obtain a normalised land system change boundary at 0.59.

| Parameter | Symbol | Value | Unit |
|---|---|---|---|
| ecosystem-dependent conversion factor | $r_{tc}$ | 2.5 | $a^{-1}$ |
| scaling factor for photosynthesis $P(T)$ | $a_p$ | 0.48 | $(20K)^{-b_P}$ |
| scaling factor for respiration $R(T)$ | $a_r$ | 0.40 | $(20K)^{-b_r}$ |
| power law exponent for increase in $P(T)$ for low $T$ | $b_p$ | 0.5 | 1 |
| power law exponent for increase in $R(T)$ for low $T$ | $b_r$ | 0.5 | 1 |
| rate of exp. decrease in $P(T)$ for high $T$ | $c_p$ | 0.556 | $(20K)^{-1}$ |
| rate of exp. decrease in $R(T)$ for high $T$ | $c_r$ | 0.833 | $(20K)^{-1}$ |
| scaling factor for terrestrial carbon carrying capacity | $a_k$ | -0.6 | 1 |
| rate of exp. increase for terrestrial carbon carrying capacity | $b_k$ | 13.0 | 1 |
| offset for terrestrial carbon carrying capacity | $c_k$ | 0.75 | 1 |
| human terrestrial carbon offtake rate | $\alpha$ | 0.0004 | $a^{-1}$ |
| | | | |
| slope of $T - C_a$ relationship | $a_T$ | 1.06 | 20K |
| intercept of $T - C_a$ relationship | $b_T$ | 0.227 | 20K |
| | | | |
| carbon solubility in sea water factor | $\beta$ | 0.654 | 1 |
| atmosphere ocean diffusion coefficient | $a_m$ | 0.0166 | 20K |
| | | | |
| (*) atmospheric carbon threshold of tCDR implementation | $\tilde{C}_a$ | $0 - 0.3$ | 1 |
| rapidity of tCDR ramp-up (tCDR implementation capacity) | $s_{CE}$ | 200 | 1 |
| (*) maximum tCDR rate | $\alpha_{max}$ | $0 - 0.03$ | $a^{-1}$ |
| | | | |
| (*) size of geological fossil carbon stock | $c_{max}$ | $0 - 0.51$ | 1 |
| industrialization rate | $r_i$ | 0.03 | $a^{-1}$ |
| | | | |
| climate change boundary | $b_a$ | 0.21 | 1 |
| land system change boundary | $b_l$ | 0.59 | 1 |
| ocean acidification boundary | $b_m$ | 0.31 | 1 |

**Table 2.** Calibrated model parameters. After normalization to preindustial carbon pools, remaining units are years (a) and temperature (20K). Parameters marked with an asterisk (*) are varied during the analysis and the parameter range is stated.



### 2.4 Model analysis and terminology

Our analysis of the co-evolutionary system aims at assessing transient dynamics of carbon pools
with respect to planetary boundaries. First (Sect. 3.1), we run the model and exemplarily show the
influence of socially controlled parameters of tCDR implementation on the transient carbon pool
evolution. It is of particular relevance under what circumstances the simulated carbon pool trajec-
tories (atmosphere, ocean and land) do not cross their respective planetary boundaries. We refer to
the regions on the safe side of the planetary boundaries as *safe regions*. All carbon pool trajectories
remaining in the respective safe region at all times are considered *safe trajectories*. For example, all
atmospheric carbon trajectories that do not cross the planetary boundary for climate change (i.e. tra-
jectories that are in the safe region of atmospheric carbon) are safe atmospheric carbon trajectories.
System states with each carbon pool remaining in its respective safe region are referred to as carbon
system states in the safe operating space, i.e. *safe states*.

In a nonlinear dynamical system, trajectories can be sensitive to initial conditions. The preindus-
trial distribution of carbon pools, as well as carbon dynamics in the Earth system are relatively well
assessed, while still subject to high uncertainty (Ciais et al., 2013). Furthermore, considerable un-
certainty remains with respect to our conceptual model structure and the exact values of planetary
boundaries. Bearing in mind these inherent uncertainties, we explore how robust the existence of
safe trajectories is under a variation of the initial conditions, i.e. the initial carbon pool distribution,
and different tCDR characteristics (Sect. 3.2).

Such a variation of initial conditions is also a common approach to conceptualising and measuring
resilience of social-ecological systems as the ability to return to an attracting state after a perturbation
(Holling, 1973; Scheffer et al., 2001). A suitable approach to quantifying the likelihood of a complex
system to return to an attracting state under finite perturbations is basin stability analysis (Menck
et al., 2013).

In the context of planetary boundaries, not necessarily all trajectories that approach a *safe attrac-
tor* (i.e. an attractor within the SOS associated to all three planetary boundaries) would be considered
safe, because they could temporarily leave the safe region. The concept of constrained basin stabil-
ity (van Kan et al., 2016) and related methods (Hellmann et al., 2015) provide generalisations of
basin stability that allow taking transient phenomena into account. Similarly to the constrained basin
stability approach, we classify different domains in state space based on transient dynamics of car-
bon pools. The set of initial conditions resulting in safe carbon trajectories form the *safe domain*.
We refer to this domain as the *manageable core of the SOS (MCSOS)*, as it depends on the tCDR
management characteristics and the emission pathway. The *undesirable domain* is formed by all
initial conditions resulting in a transgression of all three carbon boundaries at some point in time.
Remaining state space domains are formed by initial conditions leading to a transgression of a subset
of planetary boundaries. They are referred to as the respective *partially manageable domains (MD)*





(e.g. the land manageable domain is the state space domain of initial conditions with trajectories without a transgression of the land boundary).

The computational efficiency of our model allows for a systematic analysis of the MCSOS and other domains under variation of societal parameters (tCDR management and fossil fuel emissions).

We analyse how the size of all domains (MCSOS, partially MDs and the undesirable domain) varies with different tCDR characteristics (Sect. 3.3) and emission pathways (Sect. 3.4). In the spirit of van Kan et al. (2016), the size of (partially) manageable domains can be interpreted as a resilience-like measure of the opportunities to stay within the carbon related SOS, taking into account inherent structural uncertainties of our model, the location of planetary boundaries, and the preindustrial car-

bon pool distribution. Note that the maximum extent of the MCSOS is constrained by the planetary boundaries, but it may differ from the SOS (i.e. the *safe* region), as the safety of the domain is determined by transient system dynamics, whereas the SOS is defined within static planetary boundaries.

## 3 Results and Discussion

### 3.1 Carbon system trajectories subject to societal tCDR management loop

To illustrate how the co-evolutionary social-environmental system evolves with respect to carbon-related planetary boundaries, Figure 5 depicts trajectories of the major carbon pools with tCDR adhering to different management characteristics. All trajectories start at their respective normalised preindustrial state. The normalised planetary boundaries (Sect. 2.3) are indicated as dotted lines and the safe region of each boundary (refer to Sect. 2.4) is shaded in the respective colours. Variation

of tCDR characteristics reflects uncertainty about possible tCDR rates related to overall biomass harvesting potentials and societies' implementation capacities (Sect. 2.1).





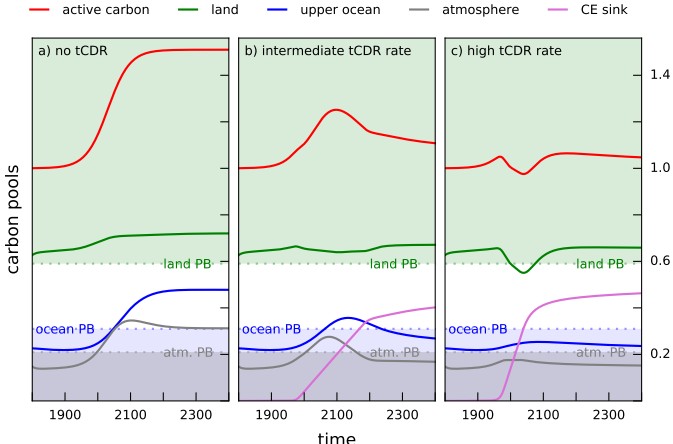

**Figure 5.** Time evolution of the normalised carbon pools in our model of the carbon system for three tCDR configurations with a high emission baseline (cumulative emissions as in RCP8.5 (Riahi et al., 2011)): a) without tCDR ($\alpha_{max} = 0$), b) intermediate tCDR rate ($\alpha_{max} = 0.0025$) and c) high tCDR rate ($\alpha_{max} = 0.025$). Total active carbon (red) is increased by fossil fuel emissions ($c_{max} = 0.51$) with dynamic response of the terrestrial carbon pool (green), maritime carbon pool (blue) and atmospheric carbon pool (grey). The tCDR sink (purple) stores carbon extracted from the active system. Shaded areas represent the respective safe regions of land, ocean and atmosphere in green, blue and grey. Dotted lines indicate the location of the associated planetary boundaries.

The emission baseline used for all results displayed in Figure 5 is a business-as-usual scenario with cumulative emissions as in RCP8.5 (Riahi et al., 2011). Without tCDR (Fig. 5a), all fossil carbon societies emit into the atmosphere is distributed to ocean, land and atmosphere. This results

in more active carbon (red), leading to carbon accumulation in all pools and a transgression of the atmosphere and ocean boundaries. In this emission scenario, the land system accumulates carbon and, thus, moves away from its planetary boundary in our model setting (note that the actual control variable of the planetary boundary of land-system change as defined by Steffen et al. (2015) is the remaining forest cover, which would not be directly modified by changing atmospheric carbon

concentrations). Moreover, higher emission baselines (results not shown here) can lead to decreasing terrestrial carbon stocks when respiration dominates over photosynthesis due to strong global warming.

In Figures 5b) and c), the societal tCDR response via harvesting from the terrestrial carbon stock and subsequent storage starts just before the atmospheric boundary is reached ($\tilde{C}_a = 0.18 \sim 340$

ppmv). With a low tCDR rate (maximal storage flux of about 7 GtC a$^{-1}$, $\alpha_{max} = 0.0025$), the CE sink is filled relatively slowly (Fig. 5b). Thus, a transient transgression of the atmosphere and ocean boundaries cannot be prevented. However, all trajectories re-enter their respective *safe* region after about 150 years. A higher tCDR rate ($\alpha_{max} = 0.025$, corresponding to very high potential storage





fluxes of 26 $\mathrm{GtC\,a^{-1}}$ or 5 % of global biomass per year) can prevent a large increase in active carbon
and thus prevents the transgression of both, atmosphere and ocean boundaries (Fig. 5c). However,
extensive harvest from the land carbon pool then leads to a temporary transgression of the land
boundary. The implementation of tCDR was thus effective in its purpose of preventing entry into a
dangerous region of climate change, but at the cost of exploiting the land system to an extent that
crossed the land system change boundary.

These results show that small tCDR rates (Fig. 5b) (or too late implementation, results not shown
here) do not necessarily keep the system in the SOS. High tCDR rates (Fig. 5c) could seem successful
when focusing on the climate change boundary, but might in fact not be feasible if other components
of the carbon system are taken into account. In light of ongoing deforestation for the purpose of
bioenergy production (Gao et al., 2011), this simulated co-transgression of the land system change
boundary with large-scale tCDR is an important and plausible feature of the model.

The carbon values stated here are primarily given as an orientation for the reader, and should
not be directly interpreted with respect to tCDR feasibility assessments. However, tCDR rates of
7 $\mathrm{GtC\,a^{-1}}$ are in line with more conservative biomass harvest potentials considering biodiversity
conservation and agricultural limits (Dornburg et al., 2010; Beringer et al., 2011). More idealistic
assessments of tCDR rates of more than 35 $\mathrm{GtC\,a^{-1}}$ – assuming high biomass yields on more than
1/4 of global land area – have been reported as well (Smeets et al., 2007). In this context, the range
of tCDR rates studied in this paper reflects both conservative and highly optimistic tCDR potentials
reported in the literature.

### 3.2 State space domain structure of the Earth's carbon system subject to societal tCDR management loop

We compute the state space domain structure (refer to Sect. 2.4) from a sample of initial conditions
around the preindustrial carbon state. We sample approximately 66,000 initial conditions from a
regular grid by variation of each carbon pool by $\pm0.2$ around the preindustrial conditions. This
range is a pragmatic choice which does not influence the following qualitative analysis. To compute
the existing domains, we evolve each initial condition for 600 years in time and colour it according
to the domains following from the transient properties of the trajectories of land, atmosphere and
ocean carbon, as described above. The mapping of initial conditions sheds light on possible domains
in the carbon system and potential transitions into other state space domains in our model of the
carbon cycle. In this context, the vicinity of the preindustrial and current Earth system states to such
domain boundaries in the model's carbon state space is of particular relevance.





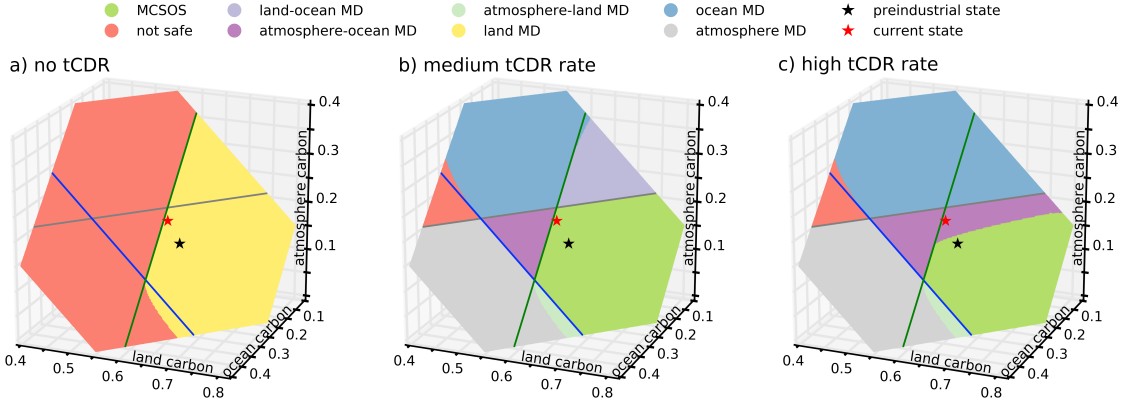

**Figure 6.** Charting of normalised carbon system state space in our model for three tCDR management characteristics with identical, relatively low emission baseline ($c_{max} = 0.2$): a) without tCDR ($\alpha_{max} = 0$), b) intermediate tCDR rates ($\alpha_{max} = 0.004$) and c) high tCDR rates ($\alpha_{max} = 0.04$). The two-dimensional plane is formed by sampling initial conditions around the preindustrial state (variation of carbon stocks by $\pm 0.2$ while conserving total carbon in the system). Each domain is coloured according to transient properties of trajectories starting in different state space regions. For example, the MCSOS (i.e. safe domain) is formed by the initial conditions of *safe* trajectories, whereas red indicates the initial conditions of trajectories crossing all respective planetary boundaries at some point of the simulation. Lines indicate the associated planetary boundaries of atmosphere, land and ocean in grey, green and blue, respectively.

Figure 6 shows the existing domains without tCDR (a), with intermediate tCDR rates (b) and with very high tCDR rates (c). The emission baseline is the same for all variations of tCDR characteristics, with cumulative emissions of approximately 880 GtC, which is comparable to RCP2.6 cumulative emissions (Vuuren et al., 2011). The current state of the carbon cycle is located in proximity to domain borders, highlighting that it is close to a transgression of the land system and climate change boundaries. Historical emissions and land system changes have moved the state of the carbon cycle closer towards the undesirable domain, and remaining on an emission trajectory similar to RCP2.6 without tCDR results in the non-existence of the MCSOS (Fig. 6a). Thus, the manageable core does not exist if the implementation of tCDR management is not considered by society, even in a relatively low emission scenario.

Figures 6b) and c) serve as an example of how human intervention and management by tCDR can influence the size and even the existence of the MCSOS and other domains. With an implementation of tCDR, the MCSOS can be re-established, potentially to its full extent, which is directly determined by the three planetary boundaries (Fig. 6b). Even for a relatively low emission scenario, the tCDR threshold needs to be at sufficiently low atmospheric carbon contents ($\tilde{C}_a = 0.16$) to prevent potential boundary transgressions. Nevertheless, because of past land use change, the current Earth system





state is approaching domains with unsafe land system and climate change. If tCDR is applied under the same conditions but with a ten times higher potential tCDR rate ($\alpha_{max}$=0.04), the MCSOS shrinks due to over-exploitation of the land system for tCDR (Fig. 6c). The current state of the

carbon cycle of the Earth system is out of the MCSOS. In this case, large societal commitment to avoid a transgression of the climate change boundary leads to a transgression of the land system change boundary in our model.

### 3.3 Size of manageable domains under variation of tCDR characteristics

The size and existence of the MCSOS and other state space domains depends on tCDR characteris-

tics (refer to Sect. 2.4). We compute the size of the different state space domains depending on the most decisive management parameters, i.e. on the implementation threshold $\tilde{C}_a$ and on the potential maximum tCDR rate $\alpha_{max}$. The size of all domains is measured in relation to the size of the considered state space section as depicted in Figure 6, which is given by a variation of preindustrial conditions by $\pm 0.2$.

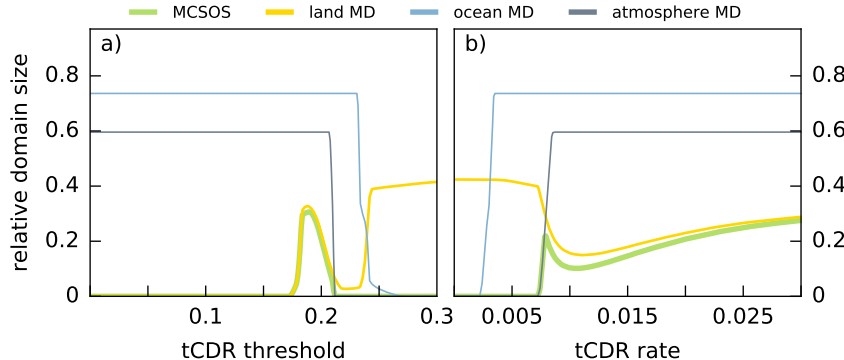

**Figure 7.** Relative size of domains in modelled carbon system state space for normalised parameter variation of a) tCDR threshold (with $\alpha_{max} = 0.02$) and b) tCDR rate (with $\tilde{C}_a = 0.2$) for a medium emission scenario ($c_{max} = 0.4 \sim 1600$ GtC cumulative emissions). All domain sizes are given as shares of the state space region defined by a variation of the preindustrial conditions by $\pm 0.2$.

Figure 7 depicts the relative size of the MCSOS and the partially manageable domains under baseline emissions of $c_{max} = 0.4$, corresponding to cumulative emissions on the order of RCP6.0. The size of the MCSOS or partially MDs can be interpreted as a form of resilience of the system (i.e. the likelihood that the system stays within the carbon related SOS). Thus, we measure the resilience of the carbon cycle by the size of MCSOS (i.e. the opportunity of success of tCDR to maintain

safe trajectories). This strongly depends on the atmospheric carbon threshold at which tCDR is implemented. Obviously, only the anticipation of an approaching planetary boundary can prevent a transgression thereof. Thresholds higher than the atmospheric carbon boundary ($b_l = 0.21$) are not





sufficient in sustaining a MCSOS, because the atmosphere MD disappears by definition at $\tilde{C}_a = 0.21$
(grey line in Fig. 7a).

However, strong anticipation coupled with too early tCDR implementation does not necessarily
maintain the system within the SOS. If tCDR is initialized at relatively low atmospheric carbon
contents ($\tilde{C}_a = 0.13$ (approx.330 ppmv) in Fig. 7a), the MCSOS is diminished due to a transgression
of the land system change boundary at some point in time. Hence, the window of opportunity for
using tCDR as a means of staying in the SOS under this exemplary fossil fuel emission scenario is
limited to a relatively narrow range of tCDR implementation thresholds.

Similar to the tCDR threshold, the parameter governing the maximal achievable rate of tCDR
plays a decisive role for the existence of the MCSOS. With a tCDR implementation threshold not far
below the atmospheric carbon boundary ($\tilde{C}_a = 0.2$), high tCDR rates are required in order to main-
tain a MCSOS. TCDR starts being effective in maintaining a MCSOS at a rate of $\alpha_{max} > 0.007$
(corresponding to approx. 16.5 GtC $a^{-1}$ with a fixed land carbon pool of 0.6). Rates smaller than
that are not sufficient because of a lacking atmospheric MD (grey line in Fig. 7b). The carbon cycle
in our model shows nonlinear behaviour for higher tCDR rates to decrease atmospheric carbon via
tCDR. Higher tCDR rates result in a smaller land MD due to over-exploitation of the photosynthetic
productivity of the system until $\alpha_{max} = 0.01$. Higher rates, however, lead to overall smaller reduc-
tions of the land MD. This nonlinearity is evoked by the co-evolutionary feedbacks between society
and the carbon cycle, which lead to a deceasing tCDR flux if the system is in the atmosphere MD.
Thus, sufficiently high tCDR rates lead to fast atmospheric carbon decrease and tCDR is switched
off before the land system boundary is transgressed.

This analysis of the size of state space domains suggests that the success of tCDR in sustaining
the Earth system's persistence in the carbon SOS nonlinearly depends on the characteristics of tCDR
implementation. On the one hand, foresightedness and anticipation of planetary boundaries are re-
quired to maintain the MCSOS, while on the other hand, too early or too intensive management
could trigger co-transgressions of other planetary boundaries.

### 3.4   Opportunities and limitations of tCDR

While anticipation and appropriate management are necessary, the underlying emission scenario
plays a major role in the resulting carbon dynamics. Figure 8 exemplarily depicts the relative MC-
SOS size for variations of tCDR characteristics (threshold and potential maximum rate) for emis-
sion pathways in accordance with RCP cumulative emission scenarios. The window of opportunity
for successful tCDR decreases with increasing emission baselines. In the case of the low emission
RCP2.6 scenario ($c_{max} = 0.2$), the MCSOS can be sustained for a broad range of parameter val-
ues (Fig. 8a). The medium emission scenarios RCP4.5 ($c_{max} = 0.31$, Thomson et al. (2011)) and
RCP6.0 ($c_{max} = 0.36$, Masui et al. (2011)) show a more narrow range of tCDR characteristics that





have the potential to sustain a MCSOS (Fig. 8b and c). In a business-as-usual RCP8.5 scenario, the room for manoeuvring to maintain a MCSOS is very small (Fig. 8d).

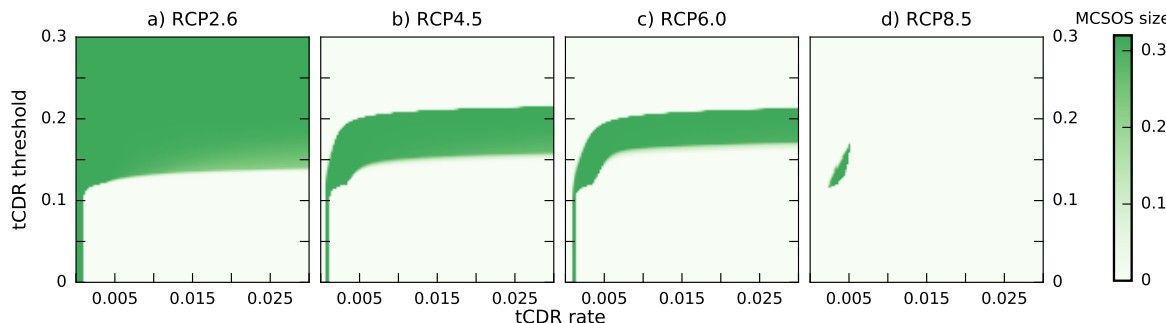

**Figure 8.** Relative size of the MCSOS for normalised parameter variation of potential maximum tCDR rate (x-axis) and tCDR threshold (y-axis) for different underlying emission scenarios: a) RCP2.6 ($c_{max} = 0.2$), b) RCP4.5 ($c_{max} = 0.31$), c) RCP6.0 ($c_{max} = 0.36$) and d) RCP8.5 ($c_{max} = 0.51$)).

This dependence of the success of tCDR on both, the tCDR characteristics and the underlying emission scenarios, highlights that any intervention into the climate system triggers a dynamic system response. In our conceptual framework, tCDR can be effective in complementing climate change mitigation strategies as employed in low emission scenarios. However, already an RCP4.5 emission scenario narrows the range of potentially successful management options significantly in compari-

son to RCP2.6 emissions. Under a business-as-usual pathway, tCDR cannot be applied to maintain a MCSOS in a resilient way. In contrast to prevailing reasoning of CE as an emergency action in case of dangerous climate change (Caldeira and Keith, 2010), tCDR would most likely not function as an emergency option under high emission scenarios when additional sustainability dimensions reflected by other planetary boundaries are taken into account.

**4 Conclusions**

Despite the fact that the reported results cannot be taken as exact quantitative prognostics of carbon pool evolution, our analysis has shown that an intervention into the climate system by climate engineering nonlinearly depends on the characteristics of terrestrial carbon dioxide removal (tCDR) implementation. TCDR for managing the atmospheric carbon pool does not necessarily safeguard

the carbon cycle in a safe operating space defined by several interlinked planetary boundaries because of potential trade-offs with land and ocean carbon pools.

The success of maintaining a manageable core of the safe operating space depends on the degree of anticipation of climate change, the potential maximum tCDR rate, as well as the underlying emission pathway. Particularly, the focus on one planetary boundary alone (e.g. climate change), may





430 lead to navigating the Earth system out of the carbon-related safe operating space due to transgression of other boundaries (e.g. land system change). This highlights the importance of an integrated sustainability assessment in the context of climate engineering (CE) and climate change mitigation via tCDR employing more advanced models. In the case of tCDR, the consequences for biosphere integrity, as well as trade-offs with agricultural land use must be taken into account among other

435 sustainability dimensions reflected by planetary boundaries and beyond.

In analogy to our analysis for tCDR, the approach followed in this paper could be transferred to other CE proposals such as ocean fertilization or solar radiation management. Additionally, it would be of interest to extend the analysis provided here and study Earth system dynamics under CE with more detailed models in line with the framework proposed by (Heitzig et al., 2016), including a full

440 topological analysis of the system with respect to the possibility of avoiding or leaving undesired domains, the reachability of desirable domains and the various management dilemmas induced by this accessibility structure.

*Author contributions.* VH and JFD designed the study. VH implemented and validated the model and performed the simulations and analysis. VH prepared the manuscript with contributions from all co-authors.

445 *Acknowledgements.* This research was performed in the context of PIK's flagship projects COPAN on Coevolutionary Pathways and OPEN on Planetary Boundaries and Opportunities. VH and WL were funded by the DFG in the context of the CE-Land project of the Priority Program "Climate Engineering: Risks, Challenges, Opportunities??" (SPP 1689). JFD thanks the Stordalen Foundation via the Planetary Boundary Research Network (PB.net) and the Earth League's EarthDoc program for financial support. The authors gratefully acknowledge

450 the European Regional Development Fund (ERDF), the German Federal Ministry of Education and Research and the Land Brandenburg for supporting this project by providing resources on the high performance computer system at the Potsdam Institute for Climate Impact Research. The authors are grateful to Jobst Heitzig, Dieter Gerten, Tim Kittel and Wolfram Barfuss for helpful comments and discussions.



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
