# Peer review of "Collateral transgression of planetary boundaries due to climate engineering by terrestrial carbon dioxide removal"

_Earth System Dynamics, 2016_

## Referee Comment (RC1) · S. Lade (Referee) · 31 May 2016

In this paper, the authors extend a stylised carbon cycle model to include climate engineering by terrestrial carbon dioxide removal (tCDR). The modelling is technically rigorous and the paper is clearly written. There are clear advances in including climate-society feedbacks in a dynamical model and in the methods of analysis.

My main suggestion is that I would like to see more concrete conclusions, for example about the likely effectivness of tCDR and/or about what insights this modelling approach achieved (or even why this modelling approach was chosen). Most of the statements in the abstract and the conclusion are rather empty (e.g. the results of tCDR depends on its parameters; there are trade-offs) or at best could also be obtained by

simple accounting of carbon stocks or emission rates.

To my mind the main advantages of a stylised dynamical model over simple carbon stock accounting are if the system under consideration has feedbacks or time lags or non-linearities that are crucial to understanding its dynamics. Perhaps this is true in the present case but I don't see it yet, at least not in your main conclusions. Can you see any consequences of dynamics on whether tCDR is likely to succeed (in keeping the earth system within, or moving it into, the SOS)? What is the time horizon on which tCDR has to start? How likely is it that tCDR will cause at least one PB to be transgressed?

Minor comments

- Line 132: It's the difference in partial pressures, not the pool size, that determines atmosphere-ocean flux

- Motivations of changes to Anderies et al.'s assumptions are clearly given, but what about discussing the validity of their assumptions and simplifications that haven't been changed? For example, the linear carbon-temperature relationship, and the single terrestrial and marine carbon stocks (which combine above and below ground and surface and deep ocean stocks, respectively).

- Lines 164-165: Why correct for carbon dioxide dynamics on long time scales? ("50% of the emitted carbon stays in the atmosphere")? Processes removing atmospheric carbon are already represented in the model. I would have thought temperature response to emissions on short time scales would have been more appropriate here. Long-time dynamics will emerge from the model.

- I realise you probably don't have control over this, but I would have preferred Table 2 at the section of section 2.2 rather than several pages later.

- Figure 4 is somewhat misleading. It suggests that the terrestrial biosphere will store carbon all the way to arbitrarily high atmospheric carbon concentrations. But in your

model, above a certain concentration the temperature will be high enough for respiration to exceed photosynthesis and you will have zero carbon storage.

- Line 205-6: Check grammar here.

- Line 215: The planetary boundary is 350ppm (Steffen, 2015). The range 350-450ppm is the 'zone of uncertainty' of the threshold at which dangerous consequences may start to happen. Therefore we have already exceeded the climate change planetary boundary, unlike what is written here and is presented in the figures.

- Line 222: Would appreciate being a little more explicit about how the number 0.31 is obtained.

- Line 225-7: I have no problem with this reasoning, but maybe be explicit about the assumptions on soil carbon. I guess the assumption is that soil carbon is unchanged by deforestation? Is this reasonable?

- Figure 8: Interesting that in (b) and (c) the parameter on the vertical axis needs to be within a narrow parameter range. Why?

- Line 423: The success of a climate intervention "nonlinearly depends" on tCDR effectiveness. This is not surprising; when the aim is to avoid a threshold (a planetary boundary), of course success will be very sensitive to parameters in the vicinity of the threshold. Or is there some other effect you're referring to?

---

## Referee Comment (RC2) · Anonymous Referee #2 · 8 Jul 2016

Review of "Collateral transgression of planetary boundaries due to climate engineering by terrestrial carbon dioxide removal" by Vera Heck, Jonathan F. Donges and Wolfgang Lucht.

General comment

I very much agree with the approach taken in this paper. We have long known that interactions among the 9 planetary boundaries (PBs) are important, but have only made qualitative assessment of these interactions so far. Applying a conceptual modelling approach to exploring a small set of PB interactions around a specific question is an excellent way to approach the interactions problem. And I fully agree that a conceptual modelling approach is an important step, as it allows one to better understand how

the model is behaving – providing insights into how the system might be operating. The outcomes of this modelling study show how effective conceptual modelling can be in elucidating system-level constraints and trade-offs in a broad sense. The authors are to be congratulating for taking such an important and convincing step forward in developing the PB framework.

Specific comments:

1. Figure 1 is an excellent visual description of the model but it leaves one interesting carbon cycle-climate question a bit unanswered. In many countries, storage of carbon in land systems via reforestation and afforestation (and avoided deforestation) is being used to "offset" fossil fuel emissions. In Figure 1, these activities would be part of the loop "Land-human offtake-land use emissions-atmosphere". These activities could be considered as "negative" human offtake, or human uptake. But the point – clearly made in Figure 1 – is that such activities clearly remain in the active carbon cycle and can in no way "offset" fossil fuel emissions. It is only when tCDR activities are undertaken, and the transfer of carbon is from Land to CE sink, can carbon originating in land truly offset emissions of carbon from the geological reservoir. Although this issue is not a part of the simulation, it might be worth including a paragraph that discusses this fundamental difference between carbon stored in above-ground vegetation (and thus in the active carbon cycle) and carbon stored in geological formations.

2. The PB for land system change is actually not based on the carbon storage on the three major forest biomes (boreal, temperate, tropical) but rather on the biogeophysical feedbacks of these three biomes to the physical climate system via changes in albedo and evapotranspiration. In the 2015 PB paper we noted that the land carbon issue, which in principle affects all terrestrial biomes (although the bulk of the above-ground biomass in land systems is in the major forest biomes), would be dealt with the climate PB, given than atmospheric $CO_2$, a feature of the active carbon cycle, was the control variable for the climate boundary. An interesting off-line calculation might be to fix the land system boundary at 75% of the carbon storage for the three major forest biomes

(based on potential areas), and then see what this means for carbon offtake for the rest of the terrestrial biosphere. This, of course, would only be interesting for those scenarios in which the land-system boundary is transgressed.

3. Just to follow on from point 2, there is an interesting further nuance to the tradeoff between the climate and land-system change PBs for very high tCDR rates – the scenarios that shrink the MCSOS due to transgression of the land-system PB in order to meet the climate PB. This may actually be counterproductive for the climate system, given that the land system PB is configured around biogeophysical feedbacks to the climate system. If these are disrupted due to transgression of the land-system PB, we may see significant changes in atmospheric circulation, monsoon systems, rainfall patterns more generally, even though the carbon aspect of the climate PB is respected via very high tCDR rates. So there is another interesting trade-off at play here!

4. The biosphere integrity PB (along with climate one of the two core PBs) was only mentioned once, I think, in the manuscript. This is OK, as it is beyond the scope of the study. However, the 2015 PB paper noted that this boundary was more likely to be a bigger constraint on the use of land systems for carbon management than the land-system PB itself (which is rather narrowly focused on biogeophysical feedbacks to climate). There isn't much that can be done yet in a modelling framework with the biosphere integrity boundary, but there are some promising approaches such as the Biodiversity Intactness Index (BII) or MSA (Mean Species Abundance) that are quantitative and could eventually be useful in conceptual modelling frameworks. So this is just a note to say "watch this space", with no action required on the present manuscript.

5. The issue of baseline emission trajectories was a bit confusing in the paper. This is especially important since, according to the conclusions section, managing an SOS depends, in addition to the anticipation of climate change and the potential maximum tCDR, on the baseline emissions pathway. For example, RCP8.5 was used early in the analysis as the emissions pathway (cf. Figure 5), but then Figure 6 switches to a low

baseline emission pathway, while Figure 7 uses an emissions baseline of ~1600 Gt C cumulative emissions. It is only when we get to Figure 8 that we see the profound importance of the baseline emission pathway for the entire analysis! I think this problem could be rather easily fixed by putting a paragraph upfront in the paper foreshadowing that different baseline emission pathways are used in various points of the paper, and that there are good reasons for this. The para could also foreshadow the important of baseline emission pathway, but that this will be dealt with near the end of the paper.

6. I think the trade-off analyses in this paper are excellent, and are certainly a strong point of the paper. Even though this is a rather simple conceptual model, it yields some fascinating tradeoffs involving anticipation and timing of actions, as well as magnitudes of interventions. In particular, I really liked the statements in lines 394-398 and 427-429. These really show the value of this approach.

―――――――――――――――――――――――――

---

## Author Comment (AC1) · 3 Aug 2016

**Response to Steven Lade (referee #1)**

*We thank Steven Lade for his constructive review which will help us to improve the manuscript! Following the procedures of Earth System Dynamics, we will take the reviewer comments into consideration in a revised manuscript as follows if the editor approves submission of a revised paper:*

**Referee**:
In this paper, the authors extend a stylised carbon cycle model to include climate engineering by terrestrial carbon dioxide removal (tCDR). The modelling is technically rigorous and the paper is clearly written. There are clear advances in including climate-society feedbacks in a dynamical model and in the methods of analysis.

My main suggestion is that I would like to see more concrete conclusions, for example about the likely effectivness of tCDR and/or about what insights this modelling approach achieved (or even why this modelling approach was chosen). Most of the statements in the abstract and the conclusion are rather empty (e.g. the results of tCDR depends on its parameters; there are trade-offs) or at best could also be obtained by simple accounting of carbon stocks or emission rates.

To my mind the main advantages of a stylised dynamical model over simple carbon stock accounting are if the system under consideration has feedbacks or time lags or non-linearities that are crucial to understanding its dynamics. Perhaps this is true in the present case but I don't see it yet, at least not in your main conclusions. Can you see any consequences of dynamics on whether tCDR is likely to succeed (in keeping the earth system within, or moving it into, the SOS)? What is the time horizon on which tCDR has to start? How likely is it that tCDR will cause at least one PB to be transgressed?

> *Reply:*
> *The transient transgression of planetary boundaries can only be simulated with a dynamic model. This is why we used the modelling approach developed by Anderies et al 2013 which was specifically developed for application in the context of planetary boundaries. We will edit the manuscript to emphasize the relevance of the societal feedback to the atmosphere (i.e. monitoring atmospheric carbon -> action thereon in form of tCDR -> atmospheric and other carbon compartments' response -> monitoring atmospheric carbon).*
>
> *Reliable assessments of the likely 'real-world' effectiveness of tCDR (and the required time horizon) are not achievable with this conceptual model. Our approach was rather meant to analyse the dynamic interaction of the societal feedback and carbon pools in a planetary boundaries context. However, our constrained basin stability based approach allows an estimation of tCDR effectiveness via the size of the manageable core of the SOS (MCSOS). In the abstract and conclusions of the manuscript, we will put more emphasis on the fact that our conceptual modelling results suggest that tCDR could only successfully be deployed as part of a strong climate change mitigation scenario and is not likely to be effective in a business-as-usual or climate emergency scenario. We will add the conclusion that in light of numerous economically based integrated assessment studies on tCDR, it is of special importance to note that societal focus on climate change only is likely to come at large costs to the biosphere.*

Minor comments

- Line 132: It's the difference in partial pressures, not the pool size, that determines atmosphere-ocean flux

> *Reply: In the model, the rate of ocean-atmosphere diffusion is approximated as being proportional to the difference in size of the atmospheric carbon pool and the maritime carbon pool. We will change Line 132 to emphasize that the dependence on carbon pool size is an assumption of the model which does not simulate carbon concentrations. (The validity of this assumption is discussed in the Appendix 4 of Anderies et al.)*

- Motivations of changes to Anderies et al.'s assumptions are clearly given, but what about discussing the validity of their assumptions and simplifications that haven't been changed? For example, the linear carbon-temperature relationship, and the single terrestrial and marine carbon stocks (which combine above and below ground and surface and deep ocean stocks, respectively).

> *Reply: In the methods section we will add information about existing simplifications of the model such as the single terrestrial and maritime carbon pools. Only the upper ocean carbon pool is included because the movement of carbon into the deep ocean occurs on longer timescales relative to those of interest, as discussed by Anderies et al. The single land carbon pool is motivated by a simple proportional partitioning of aboveground and belowground carbon pools (Anderies et al.) These simplifications have been adopted, because they reduce the number of state variables and we were able to qualitatively reproduce the dynamics of observed carbon pool evolution with the model. Within the scope of our study, the addition of two more dimensions to include above and belowground terrestrial carbon and surface and deep ocean carbon would not have been feasible.*

- Lines 164-165: Why correct for carbon dioxide dynamics on long time scales? ("50% of the emitted carbon stays in the atmosphere")? Processes removing atmospheric carbon are already represented in the model. I would have thought temperature response to emissions on short time scales would have been more appropriate here. Long-time dynamics will emerge from the model.

> *Reply: Thank you for pointing this out, the calibration was described in a misleading way. In the model the temperature is linearly dependent on atmospheric carbon content $T(C_a)$. For the calibration, however, we used the transient response of temperature to cumulative emissions (TRCE) (i.e. a linear relationship of temperature to cumulative emissions of 2K/1000GtC). We transformed this (under the assumption that 50% of emitted carbon stay in the atmosphere) to the warming rate of 2K/(500GtC in the atmosphere) for the calibration of $T(C_a)$ instead of T(cumulative emissions). We will rewrite this in the final manuscript.*

- I realise you probably don't have control over this, but I would have preferred Table 2 at the section of section 2.2 rather than several pages later.

> *Reply: We agree with the referee and would move Table 2 to the end of Section 2.2*

- Figure 4 is somewhat misleading. It suggests that the terrestrial biosphere will store carbon all the way to arbitrarily high atmospheric carbon concentrations. But in your model, above a certain concentration the temperature will be high enough for respiration to exceed photosynthesis and you will have zero carbon storage.

> *Reply: Whether the land system acts as a sink or as a source is only governed by the net flux of photosynthesis and respiration (Eq. 6). In turn, the terrestrial carbon carrying capacity (depicted Figure 4) determines the maximum capacity of the system to store carbon.*
> *We will clarify this in the manuscript.*

- Line 205-6: Check grammar here.

> *Reply: Thank you for the hint!*

- Line 215: The planetary boundary is 350ppm (Steffen, 2015). The range 350-450ppm is the 'zone of uncertainty' of the threshold at which dangerous consequences may start to happen. Therefore we have already exceeded the climate change planetary boundary, unlike what is written here and is presented in the figures.

> *Reply: For this study, we used the mean of the uncertainty range (350-450ppm) as boundary value because critical atmospheric thresholds are likely to be located somewhere within the uncertainty range. Our results are qualitatively robust with respect to choice of the threshold values. We will add to the manuscript that the actual proposed boundary is located at 350 ppm.*

- Line 222: Would appreciate being a little more explicit about how the number 0.31 is obtained.

> *Reply: Unfortunately, it is not possible to derive or approximate ocean acidification solely from maritime carbon content because it largely depends on chemical variables such as pH-value, ocean alkalinity, dissolved inorganic carbon, etc. that are not included in the model. Therefore we are only able to do a very simple estimation that the boundary is located at ocean carbon pools about 5% higher than current carbon pools in the upper ocean. However, the exact location and normalisation of the boundaries is not decisive for our results and slightly different sets of planetary boundaries would not qualitatively change the systemic effects reported in this study.*

- Line 225-7: I have no problem with this reasoning, but maybe be explicit about the assumptions on soil carbon. I guess the assumption is that soil carbon is unchanged by deforestation? Is this reasonable?

> *Reply: We will be more explicit on the assumptions in the revised manuscript. In detail (which is not represented in the model), the global land carbon pool consists of soil and vegetation carbon of both, forests and savannah, grasslands, croplands. For our calculation of the planetary boundary of land system change (allowing 25% vegetation carbon loss) on the one hand 'neglects' vegetation carbon of all other biomes than forest biomes, while at the same time neglecting soil carbon changes by deforestation (which would occur to some extend (Heck et al. 2016)). Thus, we do not assume zero soil carbon losses from deforestation but rather approximate that soil carbon losses are of the same order of magnitude as the 'neglected' vegetation carbon of non-forest biomes.*

- Figure 8: Interesting that in (b) and (c) the parameter on the vertical axis needs to be within a narrow parameter range. Why?

> *Reply: Thank you for pointing out that a discussion of this important finding was missing in this part of the manuscript. The narrow range of tCDR implementation thresholds is due to the dynamic feedbacks of the model. As explained in Section 3.3 (for a fixed tCDR rate), thresholds higher than the atmospheric carbon boundary (0.21) are not sufficient in preventing a boundary transgression in a medium emission scenario, as tCDR action would start too late to prevent a transgression. This determines the upper parameter range (around 0.21). However, 0.21 is not a clear cut-off value but tCDR thresholds allowing for the existence of the MCSOS still depend on the tCDR rate; for relatively small tCDR rates a lower threshold is required than for large tCDR rates. The lower range of the tCDR threshold can be explained by the carbon dynamics of the model. As explained in Section 3.3. the MCSOS can not be sustained if tCDR thresholds are too small because of a resulting transgression of the land system change boundary. From Fig. 8 it becomes apparent that the range of tCDR thresholds depends on the tCDR rate.*
>
> *We will discuss this in the revised manuscript.*

- Line 423: The success of a climate intervention "nonlinearly depends" on tCDR effectiveness. This is not surprising; when the aim is to avoid a threshold (a planetary boundary), of course success will be very sensitive to parameters in the vicinity of the threshold. Or is there some other effect you're referring to?

> *Reply: Yes, on the one hand there is the obvious nonlinearity in the vicinity of the planetary boundary but as explained in the reply to the previous comment there is also the nonlinear feedback of the terrestrial carbon pool transgression. We will also make this point clearer in the conclusion.*

---

## Author Comment (AC2) · 3 Aug 2016

Response to anonymous referee #2

We thank the reviewer for the constructive review! If the editor approves submission of a revised paper, we will take the reviewer comments into consideration as follows:

**Referee**:

General comment

I very much agree with the approach taken in this paper. We have long known that interactions among the 9 planetary boundaries (PBs) are important, but have only made qualitative assessment of these interactions so far. Applying a conceptual modelling approach to exploring a small set of PB interactions around a specific question is an excellent way to approach the interactions problem. And I fully agree that a conceptual modelling approach is an important step, as it allows one to better understand how he model is behaving – providing insights into how the system might be operating. The outcomes of this modelling study show how effective conceptual modelling can be in elucidating system-level constraints and trade-offs in a broad sense. The authors are to be congratulating for taking such an important and convincing step forward in developing the PB framework.

> *Reply: Thank you very much!*

Specific comments:

1. Figure 1 is an excellent visual description of the model but it leaves one interesting carbon cycle-climate question a bit unanswered. In many countries, storage of carbon in land systems via reforestation and afforestation (and avoided deforestation) is being used to "offset" fossil fuel emissions. In Figure 1, these activities would be part of the loop "Land-human offtake-land use emissions-atmosphere". These activities could be considered as "negative" human offtake, or human uptake. But the point – clearly made in Figure 1 – is that such activities clearly remain in the active carbon cycle and can in no way "offset" fossil fuel emissions. It is only when tCDR activities are undertaken, and the transfer of carbon is from Land to CE sink, can carbon originating in land truly offset emissions of carbon from the geological reservoir. Although this issue is not a part of the simulation, it might be worth including a paragraph that discusses this fundamental difference between carbon stored in above-ground vegetation (and thus in the active carbon cycle) and carbon stored in geological formations.

> *Reply: This is an excellent observation! Afforestation activities for offsetting fossil fuels would be a negative human offtake flux, i.e. adding carbon to the land system which is then included in the active carbon cycle. As this study focuses on the implications of climate engineering (not afforestation), we did not explicitly include a positive and a negative human offtake flux, rather the net flux of human offtake. We will discuss this difference between afforestation and biomass extraction into a geological reservoir in the introduction.*

2. The PB for land system change is actually not based on the carbon storage on the three major forest biomes (boreal, temperate, tropical) but rather on the biogeophysical feedbacks of these three biomes to the physical climate system via changes in albedo and evapotranspiration. In the 2015 PB paper we noted that the land carbon issue, which in principle affects all terrestrial biomes (although the bulk of the above-ground biomass in land systems is in the major forest biomes), would be dealt with the climate PB, given than atmospheric $CO_2$, a feature of the active carbon cycle, was the control variable for the climate boundary. An interesting off-line calculation might be to fix the land system boundary at 75% of the carbon storage for the three major forest biomes (based on potential

areas), and then see what this means for carbon offtake for the rest of the terrestrial biosphere. This, of course, would only be interesting for those scenarios in which the land-system boundary is transgressed.

> *Reply: Due to the lack of biogeophysical feedbacks in the model, we used the land carbon content as a proxy for deforestation by measuring the loss of vegetation carbon with deforestation. We are aware that the global land carbon pool consists of soil and vegetation carbon of both, forest and non forest biomes.  Our calculation of the planetary boundary of land system change (allowing 25% vegetation carbon loss) on the one hand 'neglects' vegetation carbon of all non-forest biomes, while at the same time neglecting soil carbon changes by deforestation (which would occur to some extend (Heck et al. 2016)). By assuming that soil carbon losses are of the same order of magnitude as the 'neglected' vegetation carbon of non-forest biomes, we implicitly accounted for the discrepancy of global and forest carbon storage.*

3. Just to follow on from point 2, there is an interesting further nuance to the tradeoff between the climate and land-system change PBs for very high tCDR rates – the scenarios that shrink the MCSOS due to transgression of the land-system PB in order to meet the climate PB. This may actually be counterproductive for the climate system, given that the land system PB is configured around biogeophysical feedbacks to the climate system. If these are disrupted due to transgression of the land-system PB, we may see significant changes in atmospheric circulation, monsoon systems, rainfall patterns more generally, even though the carbon aspect of the climate PB is respected via very high tCDR rates. So there is another interesting trade-off at play here!

> *Reply: Thank you for pointing this out. We will include this in the discussion and conclusion!*

4. The biosphere integrity PB (along with climate one of the two core PBs) was only mentioned once, I think, in the manuscript. This is OK, as it is beyond the scope of the study. However, the 2015 PB paper noted that this boundary was more likely to be a bigger constraint on the use of land systems for carbon management than the land-system PB itself (which is rather narrowly focused on biogeophysical feedbacks to climate). There isn't much that can be done yet in a modelling framework with the biosphere integrity boundary, but there are some promising approaches such as the Biodiversity Intactness Index (BII) or MSA (Mean Species Abundance) that are quantitative and could eventually be useful in conceptual modelling frameworks. So this is just a note to say "watch this space", with no action required on the present manuscript.

> *Reply: Yes, we agree that it would be very interesting to include some biodiversity index in future works.*

5. The issue of baseline emission trajectories was a bit confusing in the paper. This is especially important since, according to the conclusions section, managing an SOS depends, in addition to the anticipation of climate change and the potential maximum tCDR, on the baseline emissions pathway. For example, RCP8.5 was used early in the analysis as the emissions pathway (cf. Figure 5), but then Figure 6 switches to a low baseline emission pathway, while Figure 7 uses an emissions baseline of ~1600 Gt C cumulative emissions. It is only when we get to Figure 8 that we see the profound importance of the baseline emission pathway for the entire analysis! I think this problem could be rather easily fixed by putting a paragraph upfront in the paper foreshadowing that different baseline emission pathways are used in various points of the paper, and that there are good reasons for this. The para could also foreshadow the important of baseline emission pathway, but that this will be dealt with near the end of the paper.

> *Reply: Thank you for pointing this out! We will add a section on baseline emissions as part of the model description, foreshadowing the importance of the baseline emissions.*

6. I think the trade-off analyses in this paper are excellent, and are certainly a strong point of the paper. Even though this is a rather simple conceptual model, it yields some fascinating tradeoffs involving anticipation and timing of actions, as well as magnitudes of interventions. In particular, I really liked the statements in lines 394-398 and 427- 429. These really show the value of this approach.

> *Reply: Thank you very much!*

---

## Author Response (AR2)

**Response to the referee Report #1 and list of changes**

We thank the reviewer for pointing out that the nonlinear carbon cycle feedbacks are lacking an explanation. In the modified manuscript, we have clarified the role of nonlinear feedbacks and identified the relevant feedbacks in several results sections (Sections 3.2, 3.3 and 3.4). Further, we put more emphasis on the precise use of the word 'feedback' and replaced it by the word 'dynamics' where this was more applicable.

**List of changes (line numbers refer to the marked up manuscript version):**

- Lines 103, 124, Fig. 1 caption: societal management loop is changed to societal management feedback loop
- Fig. 1 modified to differentiate between carbon fluxes and observation (Figure caption changed accordingly)
- lines 390ff: nonlinearity in Fig. 6 is pointed out
- Lines 421ff: nonlinear dependence of land MD on tCDR threshold explained
- Lines 435ff: nonlinear dependence of land MD on tCDR rate explained
- Lines 455: dependence of the MCSOS size pointed out
- Lines 472ff: land system feedback included
- Line 490: nonlinear carbon cycle feedbacks identified

**Other minor modifications:**

Line 420: typing error corrected Line 466: tCDR range added 'state space' replaced by ' initial condition state space', if relevant References were updated Checked for British English spelling consistency Section abbreviation changed (Sect. changed to Sec.) Figure 2: wrong label (alpha\_CE) replaced by alpha\_max 
[revised manuscript text omitted]